# Rapid adaptive phenotypic change following colonization of a newly restored habitat

Camilla Lo Cascio Sætre[1], Charles Coleiro[2], Martin Austad[2], Mark Gauci[2], Glenn-Peter Sætre[1], Kjetil Lysne Voje[1] & Fabrice Eroukhmanoff[1]

Real-time observation of adaptive evolution in the wild is rare and limited to cases of marked, often anthropogenic, environmental change. Here we present the case of a small population of reed warblers (*Acrocephalus scirpaceus*) over a period of 19 years (1996–2014) after colonizing a restored wetland habitat in Malta. Our data show a population decrease in body mass, following a trajectory consistent with a population ascending an adaptive peak, a so-called Ornstein–Uhlenbeck process. We corroborate these findings with genetic and ecological data, revealing that individual survival is correlated with body mass, and more than half of the variation in mean population fitness is explained by variation in body mass. Despite a small effective population size, an adaptive response has taken place within a decade. A founder event from a large, genetically variable source population to the southern range margin of the reed warbler distribution likely facilitated this process.

[1] Centre for Ecological and Evolutionary Synthesis, Department of Biosciences, University of Oslo, P. O. Box 1066 Blindern, N-0316 Oslo, Norway. [2] BirdLife Malta, Xemxija Waterfront Apartments, Flat ½, Triq Is-Simar, Xemxija SPB 9025, Malta. Correspondence and requests for materials should be addressed to F.E. (email: fabrice.eroukhmanoff@ibv.uio.no).

Contemporary evolution, or evolutionary changes observable over less than a few hundred generations, has been documented for a variety of species[1,2]. The threespine stickleback (*Gasterosteus aculeatus*), for example, has shown extraordinary abilities to undergo adaptive evolution within a few decades following natural colonization of novel habitats[3,4]. Evolutionary changes are often associated with shifts in the environment, as changes in adaptive optima cause new selective pressures to operate, resulting in phenotypic evolution[5,6]. These shifts in the environment are often brought on by climate change and various human activities, which have detrimental effects on many species[7,8]. Contemporary evolution is therefore often connected with conservation biology[1,9]. Many different conservation strategies are put to use, such as habitat restoration and assisted colonization, though they may not always work as well as intended[10]. When a population relocates to a new habitat, which may constitute a shift from its geographic range, new selection pressures may cause unforeseen evolutionary changes. One of the biggest challenges for populations colonizing new or restored habitat is their relatively low level of genetic diversity[11]. When effective population sizes are small, genetic variation is rapidly lost due to high rates of genetic drift, and the risk of inbreeding increases[12]. Loss of genetic variation may in return constrain a population from adapting to changes in the environment, increasing the risk of extinction[13–15]. The extent to which natural populations are able to colonize and rapidly adapt to novel habitats in connection with conservation projects has rarely been studied[16].

In this study, we follow a population of reed warblers (*Acrocephalus scirpaceus*) in Is-Simar nature reserve (Fig. 1a) in Malta from 1996 to 2014. Is-Simar was originally a waterlogged marsh, but was drained for agricultural purposes, and later used as a dump. In 1992, it became a special protection area belonging to the NATURA 2000 network and Birdlife Malta began transforming it into a nature reserve of 0.58 km² by restoring this lost wetland habitat. Is-Simar currently contains the highest number of *Phragmites* reed beds of the Maltese islands, in addition to tamarisk groves (*Tamarix* sp.), which are both suitable breeding habitats for reed warblers (Fig. 1b). Indeed, it was rapidly visited by numerous reed warbler migrants, and within 2 years, they had established a small breeding population. We show that the population has adapted to this restored habitat very rapidly, despite a small population size. There has been a decrease in body mass in the population, which correlates with an increase in mean population fitness and higher individual survival. Our results have important implications for conservation biology and evolution, especially regarding the potential success of habitat restoration in relation to a species' ability to rapidly adapt to a new environment.

## Results

**Time-series analysis.** Through the course of 19 years, the population has gone through a decrease in body mass consistent with a model of adaptive evolution (Fig. 1c). The adaptive nature of this trend is strongly supported, as it fits an Ornstein–Uhlenbeck (OU) process[17] that outperforms the neutral model of an unbiased random walk by several Akaike Information Criterion (AICc) units (Table 1). The relative support for the neutral model with genetic drift as an evolutionary driver is correspondingly weak (<1%). An OU process models how a trait evolves towards a new optimum; the trait shows directional change in the beginning of the time series as the population ascends the adaptive peak, followed by a 'stationary phase' where the trait is subjected to stabilizing selection. The trait fluctuates around the optimum due to genetic drift, plasticity and

unmeasured direct and indirect selective forces acting on the trait. The observed evolutionary trajectory of body mass follows the expected pattern of a population ascending an adaptive peak (Fig. 1c). Initially, the decrease in mean body mass is substantial, but the changes become progressively smaller and more non-directional when the population is close to the new optimum.

The rate of evolutionary change that occurred before the population reached the adaptive optimum is substantial (1.478 haldanes over the first 6 years). This corresponds to a very rapid change, but it is within the normal range of evolutionary rates measured in populations affected by human-induced environmental changes[9]. The alpha parameter ($\alpha$) from the OU model (Table 1) represents the strength of the restraining force around the optimum[6]. This parameter can be used to express the phylogenetic half-life (ln (2)/$\alpha$), which is the expected amount of time it takes for the population to evolve halfway from the ancestral state to the optimal phenotype[18]. In our case, the phylogenetic half-life is 1.76, meaning that the population is estimated to have evolved halfway to the optimal body mass in >2 years. This is, as far as we know, the fastest rate of adaptation ever recorded using time series data.

The estimated adaptive optimum was reached after ~7 years (Fig. 1c). This gradual change in the initial phase of directional selection followed by stabilization around the new optimum suggests that the population has undergone adaptive evolution, as this pattern would not be expected by adaptive plasticity alone. If the trait change were solely due to plasticity, the new population would immediately have reached the new optimal trait value, as the reaction norm for the trait in the founding population would have had to cover the optimal trait value. However, without direct evidence of heritability, the heritable and environmental components of phenotypic change cannot be partitioned definitively. While some studies on contemporary evolution have found evidence for adaptive evolution[19,20], others have pointed towards an important role of plasticity[21]. However, we are not aware of any studies that have found evidence for phenotypic changes of a plastic nature that would fit an OU model as is the case here.

**Ecological and molecular data analysis.** We further corroborate our results with analyses of field observations and mark-recapture data. The yearly estimates of mean population fitness (proportion of breeding adults) increased as the population approached the adaptive optimum (Fig. 2a). More than 58% of the variation in mean population fitness was explained by variation in body mass (Fig. 2a). The mean annual variation in body mass decreased as mean population fitness increased, suggesting that body mass has been under strong selection in this population (Supplementary Fig. 1). Also, individuals with a lower body mass survive better than heavier individuals, as can be seen from mark-recapture data (Fig. 2b). Recaptured individuals were much closer to the adaptive optimum of body mass than individuals not recaptured, the latter group being well above the optimal value. These results confirm that the decrease in body mass is an adaptation to the Maltese environment, and that this evolutionary change has resulted in a more successful breeding population. Body mass remained significantly correlated with recapture rate throughout the years (analysis of variance (ANOVA): $P = 0.007$), and there was no significant year effect (ANOVA: $P = 0.588$). Hence, selection remains fairly constant over time, despite slowing evolutionary rates.

We estimated an average mean-standardized selection gradient across years for body mass to be equal to $-0.39$ (linear regression: $P = 0.006$), which is consistent with a significant but moderate amount of directional selection on body mass[22]. Body mass has been found to be significantly heritable in the great reed warbler (*Acrocephalus arundinaceus*) as in many other passerine

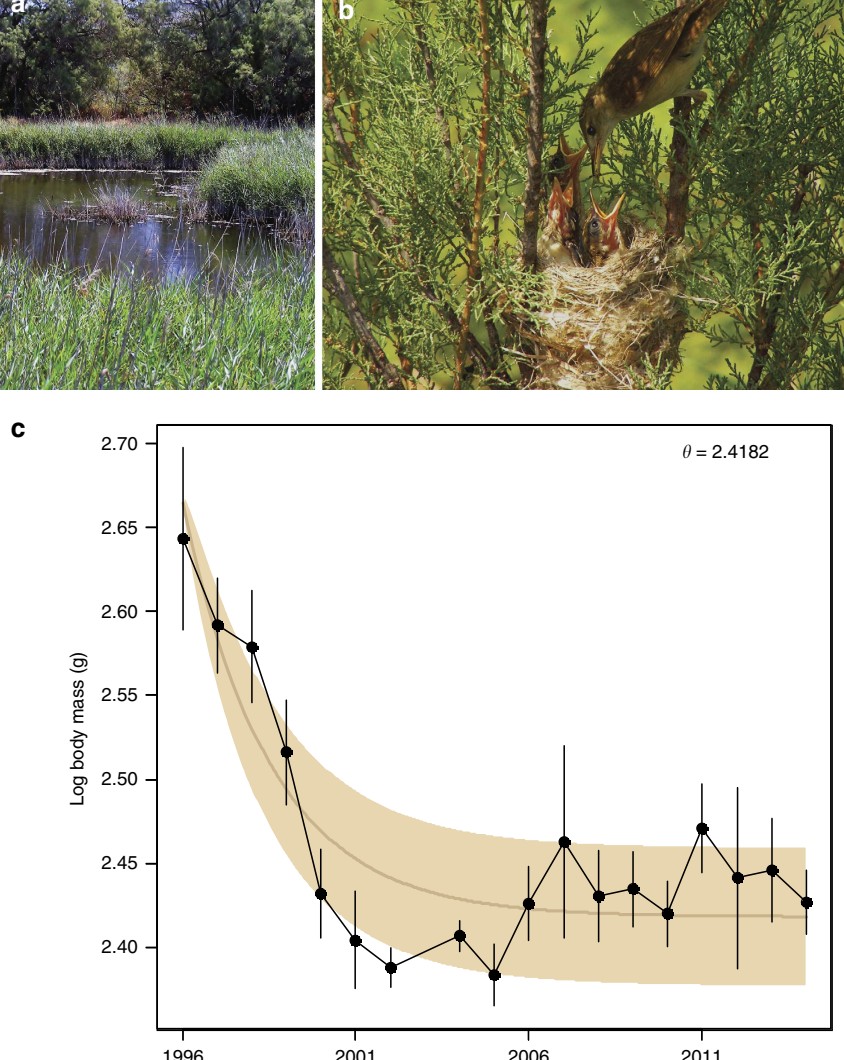

**Figure 1 | Evolution of body mass towards an adaptive optimum.** (**a**) The Is-Simar nature reserve is situated on the island of Malta, in the Mediterranean Sea. Restoration of the wetland began in 1992, where a network of pools, canals and islands were created and vegetation replanted. Shortly after, in 1994, it was colonized by reed warblers (*Acrocephalus scirpaceus*). (**b**) Reed warblers nest in reed beds (*Phragmites*) or *Tamarix* trees (almost exclusively in the latter at Is-Simar) and usually lay three to five eggs, which are incubated by both parents. (**c**) The evolution of log body mass over time (years; $N = 392$). Vertical error bars signify one standard error. The expected evolutionary trajectory of the best-fit adaptive model (OU) is shown as a line, with a 95% probability interval around in brown. The adaptive optimum ($\theta$) for log body mass is 2.42. No samples available for year 2003.

**Table 1 | Estimates of model fit for a neutral and an adaptive model of evolution for mean body mass.**

| Trait | Model | logL | K | AICc | Akaike weights | LRT |
|---|---|---|---|---|---|---|
| Body mass | Neutral | 30.45 | 2 | − 56.10 | 0.002 | |
| | Adaptive | 36.73 | 4 | − 62.38 | 0.998 | 12.55, $P = 0.002$ |

Neutral evolution was modelled as an unbiased random walk, and adaptive evolution was modelled as an Ornstein–Uhlenbeck (OU) process. For the OU model, the adaptive optimum ($\theta$) for log body mass is 2.42, the step variance ($\sigma^2_{step}$) is 0.0004 and the alpha ($\alpha$), the strength of the restraining force around the optimum, is 0.39. The log-likelihood (logL), number of parameters (K), bias-corrected Akaike Information Criterion (AICc) and Akaike weights suggest that the adaptive model is the more likely model. A likelihood ratio test (LRT), which tests the significance of the improved fit of the adaptive over the neutral model, with the latter treated as the null model, confirmed that indeed the observed changes in body mass are of an adaptive nature. The LRT statistic is distributed as a $\chi^2$, with two degrees of freedom.

bird species[23], and to be a highly evolvable trait[23]. In our data set, annual mid-parent and offspring body mass is significantly correlated irrespective of annual fluctuations in body mass, which supports the assumption that body mass has a significant heritable component in our population (Supplementary Fig. 2). Migrants presumably coming from Italy[24] have an average body mass that is very close to the initial body mass of the Maltese population in 1996 (Supplementary Fig. 3). This suggests that evolution of body mass has occurred *in situ* in Malta and is not the result of biased immigration.

Assuming that the fluctuations in body mass during the stationary phase are caused by genetic drift, we can estimate effective population size ($N_e$) from the OU model. This $N_e$ estimate represents an important investigation of the reliability of

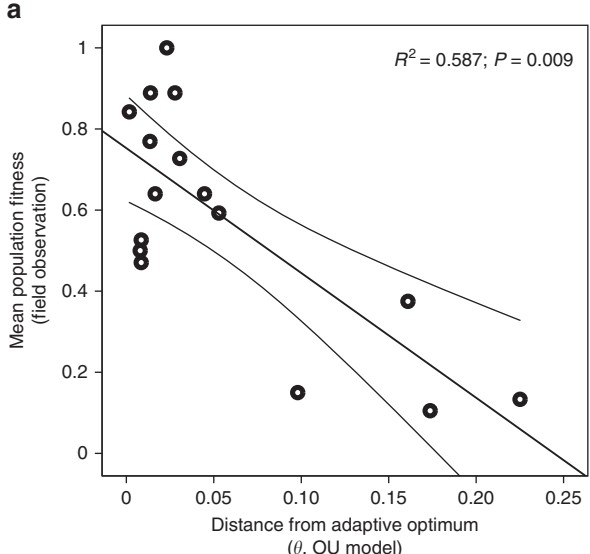

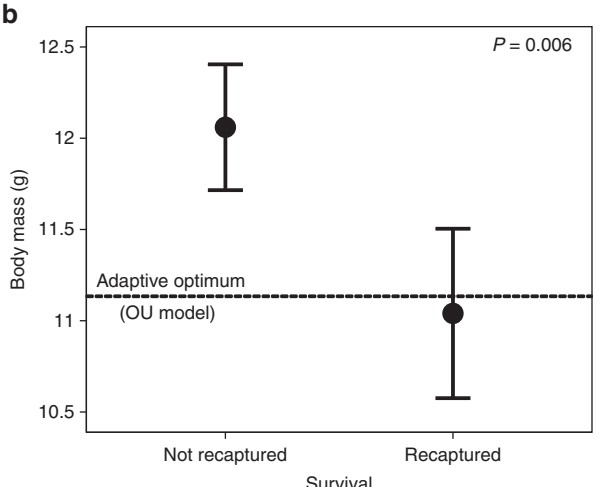

**Figure 2 | Fitness and body mass in the Maltese reed warbler population.**
(**a**) Linear regression of mean population fitness (proportion of breeding
adults each year ($N$ years $=16$)) against the yearly distance from the
adaptive optimum for body mass ($\theta$) estimated from the OU model. As the
population evolved towards the optimum, the mean population fitness
increased significantly (linear regression: $P = 0.009$) and overall, 58.7% of
the variation in mean population fitness can be explained by variation in the
distance from the adaptive optimum for body mass. (**b**) The effect of body
mass on survival. Individuals marked and recaptured have a significantly
lower body mass than individuals that are marked but never recaptured
(total $N = 198$). The dashed line corresponds to the adaptive optimum ($\theta$)
for body mass estimated from the OU model. The mean-standardized
selection gradient is equal to $-0.39$ (linear regression: $P = 0.006$).

the model as it can be compared with an independent molecular
estimate of $N_e$ (Table 2). Both estimates of $N_e$ are very similar,
and also close to an ecological estimate of $N_e$ (based on the
number of breeding pairs observed in the field). Such a consistent
match suggests that the OU model portrays the adaptive
evolution of body mass in an accurate way.

## Discussion

We suggest that reed warblers with a lower body mass could be
better suited for the hot and dry Maltese environment. Body mass
should decrease with increasing temperature[25–29], in accordance

**Table 2 | Estimates of effective populations size ($N_e$) from three independent methods.**

| $N_e$ (field) | $N_e$ (OU) | $N_e$ (molecular) | $F_{IS}$ |
|---|---|---|---|
| 7.75 (2–16) | 23.39 (4.68–32.75) | 23.60 (14.10–51) | 0.123 ($P = 0.01$) |

Field, harmonic mean of number of breeding pairs with the range shown in parentheses.
OU, $Ne1/4h^2s^2$ p/$s^2$ step, where $h^2$ is the trait heritability (set to 0.5, with 0.1–0.7 shown in
parentheses), $s^2$ p is the phenotypic variance of the sample and $s^2$ step is the step variance.
molecular, linkage disequilibrium method with $N_e$ estimator with 95% confidence intervals
shown in parentheses.

with Bergmann's rule. Lower body mass has been interpreted as an
adaptation to warmer climates, as the relatively larger body surface
areas of smaller individuals serve as efficient heat dissipators[27].
Smaller-bodied individuals are able to retain water at higher
temperatures than large-bodied individuals[30]. Other ecological
factors linked to climatic variation, such as resource availability,
may also affect body size[31]. However, rising temperatures have
previously been shown to correlate with a decrease in body
mass[19,32], which strongly point towards a thermoregulation
hypothesis. The reed warbler has been singled out as a species
that is very sensitive to global warming, which has resulted in a
recent range shift northwards[31]. Malta is the southernmost
location within the European reed warbler distribution[33], and it
has a dry subtropical Mediterranean climate, which is likely to have
imposed a strong selective pressure on colonizers.

Is-Simar, the reed warbler's main breeding site in Malta, is
small (5.8 hectares), meaning that the maximum number of
breeding pairs is constrained (5–8 pairs). However, the
probability of recruitment is high, with an estimated 10.6% of
ringed nestlings recaptured as breeding adults over the 19 years of
our study. This high recruitment rate has probably played an
important role in the adaptive process, especially in the early
stages of the colonization event. The founder population on
Is-Simar most likely consisted of individuals from a large,
genetically variable source population in Southern Europe.

With small effective population sizes, the possibilities for
adaptation should be constrained[15] and directional selection is
expected to deplete genetic variation. However, this population
has managed to reach its adaptive optimum very rapidly, despite
its small size. The $F_{IS}$-value from Table 2 signifies that there are
currently 12% fewer heterozygotes than expected under Hardy–
Weinberg equilibrium, indicating that the population is inbred.
This value is well above the average $F_{IS}$ for other European
populations ($F_{IS}$ Malta $= 0.123$, average $F_{IS}$ in Europe[33] ($N$
populations $= 31$) $= 0.05$ (s.d. $= 0.04$), $t = -10.31$, $P < 0.001$).
This may have other detrimental effects for the population in
the future, as inbreeding, for the same set of microsatellite loci in
other warbler species, has been shown to be correlated with
fitness[34,35] (reproductive output).

Furthermore, after correcting for variation due to sampling
error in trait means after the population has reached the
optimum, only $\sim 13\%$ of the stationary variance in body mass
is left unexplained (observed stationary variance $= 0.000692$,
corrected stationary variance ($\pm$ S.E.) $= 0.000092 \pm 0.00014$). This
may indicate very strong stabilizing selection for the optimal body
size in the stationary phase with little effect of drift or plasticity,
an interpretation supported by the very rapid rate of adaptation
(half-life) estimated in this system. Another possibility is that the
optimum is non-stationary, which means that part of the trait
fluctuations during the stationary phase shows how the popula-
tion is tracking the optimum's movement across years. However,
to our knowledge, the environmental conditions in terms of
habitat and resources have remained stable since the restoration
was completed in 1994, with relatively few competitors or

predators[24] and a relatively stable climate over the years[24]. Since we do not possess any pedigree data that would have allowed for a quantitative genetic assessment of the changes observed here, we cannot exclude a possible role of plasticity. Although an adaptive plastic process alone is unlikely to generate a reaction norm consistent with an OU model, we acknowledge that at least part of the changes we report here may be of a plastic nature.

The population has so far shown high potential for adaptation, but increased inbreeding could reduce the adaptability of the population. Inbreeding may amplify tendencies to deviate from the optimum and lead to maladaptation, which would threaten the future of this minuscule but evolutionary successful population. We therefore recommend further restoration of this wetland habitat allowing for a larger population, thus decreasing the risk of inbreeding depression[36] and the impact of genetic drift.

Our study may represent one of the most rapid cases of adaptive evolution ever documented in the context of habitat restoration. It also demonstrates the importance of population monitoring in evolutionary and conservation biology, as the success of a conservation project may be difficult to predict and depend largely on the evolutionary potential of the focal population or species.

## Methods

**Sampling.** From 1996 to 2014 (except for 2003 where data is not available), we sampled a total of 392 adult reed warblers during the breeding season (May–August), as part of the BirdLife Malta project. All birds included in the main analyses (unless otherwise stated) were resident individuals captured during the breeding season when no migration occurs. Birds were ringed with unique ID rings and body mass recorded to nearest 0.1 g using a digital scale. All measurements took place during the morning hours between 06:00 and 10:00 to minimize daily variation in body mass. We estimated the minimum number of breeding pairs through the intensive monitoring of nests and other frequent field observations during the entire breeding season. We also captured migrants stopping over for several days in the population during autumn migration (September–October) to investigate differences in body mass with local residents. All sampling and handling of birds was in compliance with ethical regulations, and permits for sampling were obtained from the local authorities (BirdLife Malta).

**DNA extraction.** In 2014, we sampled blood from 18 individuals. DNA was extracted from the blood samples using DNeasy Blood & Tissue kit (Qiagen), and subsequently genotyped. We amplified eight microsatellites previously used for reed warblers[33]; Aar4, Aar5, Aar8, Ase34, Ase58, Pca3, Pdoμ1 and POCC2 (Supplementary Table 1).

**Statistical analyses.** To investigate evolutionary changes through our time series of 19 years ($N = 392$), we used the average body mass of individuals caught in each year, along with corresponding sample sizes and standard deviations. If individuals were captured and measured more than once, we used the average across measurements. The average difference between two measurements at two different time points was non-significant (ANOVA: $-0.0268$ g; df = 204; $P = 0.765$). We compared the goodness of fit of a neutral (unbiased random walk) and an adaptive (OU) model to our data using the PaleoTS package[6] in R. We used bias-corrected AICc as a measure of model fit, and to show the relative support for the two models we used Akaike weights (transformations of the AICc scores to make them sum to one). We also conducted a log-likelihood ratio test using the log-likelihood estimates from the models.

To investigate the relationship between mean population fitness (proportion of breeding adults each year) and body mass, we conducted a linear regression analysis. We plotted mean population fitness to the population distance from the adaptive optimum of body mass (estimated from the OU model). We also estimated the correlation between s.d. in body mass and mean population fitness. Both these estimates should be expected to be negative if natural selection is acting on body mass. From recapture data over the entire study period ($N = 198$), we distinguished between individuals recaptured after a minimum of 21 days (but also taking in account individuals recaptured the following seasons) and individuals never recaptured, and used this as a proxy for individual survival, which we acknowledge could also be partially affected by other factors such as emigration, although our estimates on recruitment rate suggest very high philopatry. We subsequently estimated the mean-standardized selection gradient for body mass. To investigate whether there is any temporal variation in selection during the study period, we conducted a linear model of recapture probability where both body mass and year of capture were included as covariates.

We calculated different estimates of $N_e$. From field data, we estimated $N_e$ using the harmonic mean of the number of breeding pairs observed in the field across years. We further followed the procedure as described by Hunt et al.[6] to estimate $N_e$ using the parameter estimates from the OU model. $N_e = h^2 \sigma^2_p / \sigma^2_{step}$, where $h^2$ is the trait heritability, $\sigma^2_p$ is the phenotypic variance of the samples and $\sigma^2_{step}$ is the step variance, which is estimated from the model fit. We solved the equation with three different values of $h^2$ (0.1, 0.5 and 0.7). Since we do not have pedigree-based information to calculate heritability, we estimated the correlation between annual mid-parent and offspring body mass to assess the plausibility of a significant additive component of genetic variation for body mass, irrespective of environmental variation across years. Finally, we used NeEstimator (v2.01)[37] to estimate $N_e$ from our molecular data, using the linkage disequilibrium method[38].

**Data availability.** Data are available from the Dryad Digital Repository: http://dx.doi.org/10.5061/dryad.hj30r.

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

## Acknowledgements

We wish to thank A. Runemark for her help during the fieldwork and also numerous field assistants from the Maltese Ornithological Society and BirdLife Malta for their help during habitat restoration and with the subsequent monitoring of this reed warbler population. This study was funded by BirdLife Malta, the Research Council of Norway (G.-P.S.), the Nansen Foundation (F.E.) and the Faculty of Mathematics and Natural Sciences, University of Oslo (F.E.).

## Author contributions

C.C., M.A. and M.G. monitored the population and collected the morphological data. G.-P.S and F.E. designed the study and collected the genetic data. C.L.C.S. performed the DNA extractions and the analysis of the microsatellite data. C.L.C.S., K.L.V., G.-P.S. and F.E. analysed the morphological and mark-recapture data. C.L.C.S. wrote the paper with contributions and comments from all authors.

## Additional information

**Competing financial interests:** The authors declare no competing financial interests.

**Publisher's note**: 

