## [Peer Review File · Nature Communications]

Reviewers' comments:

Reviewer #1 (Remarks to the Author):

In this well-written manuscript, the authors suggest that evolutionary responses are still possible even when the population is extremely small. If this were the case, the contribution would be certainly important. Although the study is well-done in many aspects, I have a number of concerns that need to be addressed before drawing such a strong conclusion.

My major concern is that while the study provides evidence for a sorting process (i.e. natural selection), to infer that the observed phenotypic changes reflect an evolutionary response requires heritability estimates. Otherwise the observed pattern could merely reflect plastic adjustments. Thus, the authors should estimate the heritability of body mass comparing the resemblance of parents and offspring with an animal model approach or other sound approach, and then assess the extent to which the estimated evolutionary response fits with the observed changes in body mass. I also wonder why the authors focused on variation in body size, which is known to have an important environmental component, rather than on size-dependent morphological traits known to have higher heritability, like the tarsus.

A second concern is the use of the proportion of breeders as a measure of mean population fitness. While this can be criticized, it also suggests an alternative interpretation of the results: density-dependence leading to individuals in poor condition or poorly developed. Note that this would also be well-described by an OU evolutionary model. The authors can argue that the finding that recaptured individuals tended to have smaller body mass goes against this interpretation. Given that the authors examine apparent rather than actual survival, however, this later finding can reflect that the habitat is sub-optimal (the main breeding site only contained 5-8 pairs) and hence is avoided by dominant, larger individuals. If they disagree with these interpretations, the authors should discuss why these alternatives are unlikely.

Finally, I miss a convincing functional explanation linking body mass with fitness. The surface-volume ratio seems unconvincing to me because I'm not aware of any study demonstrating that this ratio affects survival in sub-tropical environments. But I could be wrong here. Please add evidence in birds suggesting that this can be an explanation for the observed decline in body size.

In sum, I think the authors have made an excellent job in conducting the analyses yet they need to provide additional evidence to convince the reader that what they report is an unprecedented evolutionary response in an extremely small population. If they can do so, the study will be an important contribution to evolutionary biology, with important implications for biological invasions (the role of evolution in invasion success is often underestimated) and conservation biology (as demonstration of evolutionary rescue).

Reviewer #2 (Remarks to the Author):

This manuscript describes the evolution of a very small population of reed warblers (*Acrocephalus scirpaceus*) following the colonization of a newly created wetland. The authors elegantly and clearly demonstrate evolution in body size using multiple approaches over only two decades after colonization. They further support a clear Ornstein-Uhlenbeck process, where the population shifts to a new adaptive peak and then shows variation around the new optimum. The authors claim that the results are particularly important given the relatively rapid rates they demonstrate in a population of less than 50 individuals (with an appreciable level of inbreeding), and they briefly discuss the implications for conservation biology more broadly.

Overall, this is a very interesting study with valuable results and implications for the broader field of evolutionary conservation biology. Contemporary evolutionary processes have only begun to be considered seriously for conservation within the last decade, and empirical results like this one are key for both understanding the implications of contemporary evolution and in demonstrating the need of conservation practitioners to consider it.

I have a few reservations about the interpretations of the result and the overall pitch of the manuscript, however:

1. The link to restoration is unclear (and probably unnecessary). Can you justify why restored habitats are an evolutionary challenge to species of conservation concern? The stated purpose of restoration is to assist population numbers and vital rates of native flora and fauna, so it is unclear why these environments would be a challenge to species due to strong changes in selective forces. I think there is a case to be made (e.g., restoration focusing on ecosystem traits that attract colonists, while not maintaining the correlations with ecosystem services that those habitat characteristics usually indicate), but it is not clear as written, and seems, in fact, opposite what one would expect. Further the hypothesis you propose to explain the change in selection pressure (thermoregulation) is one related to large-scale geographic location, not to anything to do with the restoration. As such, it is very unclear to me how the restoration itself is related to the larger story you are trying to tell. I think it might be simpler to merely frame the entire manuscript as "range expansion" or "colonizing a novel patch".

2. The definition of survival may be problematic. How many of the recaptures were on the same year versus a subsequent breeding season? How high is survival on the breeding grounds? Presumably the majority of mortality occurs during migration in this species, so by defining mortality the way you have here you have dramatically increased the probability that the "not recaptured" individuals are actually passage birds or floaters that never set up stable territories on Malta. In effect then your comparison between "recaptured" and "not recaptured" becomes a comparison between birds in the population and outside of it.

3. All of the results can be explained as evolution due to immigration and emigration. It

seems possible that the evolutionary change you document here could be the result of biased colonization by body size. This colonization bias could occur in conjunction with selection at the site (e.g., small bodied birds have higher fitness on site, but are less likely to get there due to correlations between body size and dispersal rate, so they fill in and outcompete larger bodied birds later), but colonization bias alone could also explain the results (e.g., the distribution of birds in the stable phase of the O-U process more closely represent the distribution of birds from the regional colonizing pool, but it took time to recreate that distribution because there are biases in colonizing potential that are associated with body size - larger bodied birds still get there first). This latter potential mechanism seems to be supported by the apparent increases in body size variance during the stable phase (doubly so, since the population size is actually larger at this point and that should make the SE appear to do the opposite?). But if you were to report data on the distribution of body sizes elsewhere in southern Europe, this might assist in this regard. Your inbreeding coefficient gives some evidence to the contrary (i.e., if the population has a heterozygote deficit, that is unlikely to be produced entirely by immigration, assuming that the source populations are larger than this one ... which is reasonable), but it is hard to reject the fact that inbreeding might be much higher if it were not for new immigrants (can you calculate expected FIS for a population of that size and see how far off you are?). This is a very small population with high juvenile return rates. Further, your effective population size is three times the size estimated via field observations, which argues that both your genetic and evolution models are sampling part of a larger metapopulation that includes immigrants (given the size of the wetland, would it be fair to say that it is unlikely there could have been three times as many birds with stable territories on average each year than what you detected?). Do you have good rates of immigration? Your observation on the speed of colonization would suggest that it is potentially high. Regardless, and most importantly, the fact that evolution is being driven by imm/emigration processes and not selection on site does not negate your findings in any way. It can still be measured as "selection", it is just selection due to colonization bias. The same O-U evolutionary processes can be invoked. It's a range-edge phenomenon vaguely akin to what has been reported in cane toads in Australia and holm oak in Europe. I just think it is critically important that you discuss how it is not possible to differentiate between the two possible pathways. Currently much of your language implies on-site selection.

More minor comments

Ln 106 Can you estimate this over different time frames to see if selection is consistent despite slowing evolutionary rates?

Ln 112 Please admit that many other mechanisms are possible

Ln 116 Under your model the fluctuations should be a combination of drift and stabilizing selection, if you are really at an "optimum". Given the SE you show in Fig. 1, it seems as though the population spans the projected "optimum" each generation, so I have no issue with assuming neutral net selection on average and looking for drift (although it would be good to be explicit about this assumption, as it sounds like you are claiming there is actually no selection at that point) to calculate the effective population size. But more interestingly, can you detect the reversed direction of selection if you treat the population as two, those below and those above the optima? Given the strong selection you show in the "approaching

limb" phase, and the strength of the peak attractor, it should be detectable, even if you have to pool across time steps to obtain sufficiently large populations on either side of the optimum during this phase.

Ln 160 Were the birds individually marked with unique color combinations? I'm assuming so, but it would be good to mention.

Ln 346 I think you should explain here how you defined survival, but see my comment above, I think this definition of survival may be problematic

Reviewer #3 (Remarks to the Author):

This manuscript documents rapid phenotypic change in a small population of reed warblers following the colonization of a restored marshland. Over a 19-year period, the mean body mass in this population declined, and variation in fitness was correlated with body mass, suggesting the observed changes were adaptive. Because the phenotypic changes exhibit an initially rapid decline, and are then followed by a more stationary phase, the interpretation is that the pattern of change is consistent with an Ornstein-Uhlenbeck process of how a trait evolves towards a new optimum.

I was very excited to see these results in the context of a restoration project, as it places evolutionary biology within a conservation framework. Unfortunately, the conclusion that the observed phenotypic changes represent evolution are not supported. The study fails to directly test what proportion of the phenotypic variation is due to heritable vs. environmental causes. Numerous other studies have documented similar patterns of phenotypic change in natural populations, but they have also used pedigree-based animal models to test for heritable changes. For example, Charmantier et al. 2008 *Science* documented similar rapid phenotypic change over a 47-year period, but concluded the change were due to adaptive plasticity, not evolution. Indeed, the results presented in Charmantier et al. 2008 are qualitatively very similar to those presented here (i.e. a rapid change followed by a leveling off). Furthermore, reviews of studies documenting contemporary phenotypic changes in natural populations conclude that most of the observed changes also reflect plasticity rather than evolutionary change (e.g. Hendry et al. 2008 *Molecular Ecology*; Merilä and Hendry 2014 *Evolutionary Applications*).

My suggestion would be to revise the interpretation of the results to acknowledge that the observed changes could be due to either evolution or plasticity. Presenting the results as adaptive phenotypic change, rather than adaptive evolution, does not make the observed changes any less dramatic or interesting. I would also like to see a more thorough discussion of the mechanisms that could drive changes in body mass. The manuscript suggests temperature may be the mechanism, but mass, as opposed to size, can be influenced by food availability. Lastly, from a conservation perspective, a discussion of how the limits to adaptive plasticity vs. adaptive evolution might impact the long-term viability of this population would be very useful.

Response to specific comments:

Reviewer #1 (Remarks to the Author):

Reviewer's comments:

In this well-written manuscript, the authors suggest that evolutionary responses are still possible even when the population is extremely small. If this were the case, the contribution would be certainly important. Although the study is well-done in many aspects, I have a number of concerns that need to be addressed before drawing such a strong conclusion.

Response:

We thank the reviewer for recognizing the importance of our findings. We believe that we were able to address all his/her concerns, and that our conclusions are now stronger on a number of points.

Reviewer's comments:

My major concern is that while the study provides evidence for a sorting process (i.e. natural selection), to infer that the observed phenotypic changes reflect an evolutionary response requires heritability estimates. Otherwise the observed pattern could merely reflect plastic adjustments. Thus, the authors should estimate the heritability of body mass comparing the resemblance of parents and offspring with an animal model approach or other sound approach, and then assess the extent to which the estimated evolutionary response fits with the observed changes in body mass.

Response:

We agree with the reviewer that our study provides ample evidence for a sorting process linked to natural selection. However, we disagree that our results are equally consistent with a plastic adjustment.

First, one does not necessarily need heritability estimates to infer evolution, as a number of evolutionary frameworks rely on other tools (e.g. the comparative method). Although we do not have pedigree data that would allow us to directly demonstrate the genetic basis to the changes we describe, we have several lines of evidence which support adaptive evolution. The mathematical modelling approach we have used on our time series data strongly supports that an Ornstein-Uhlenbeck (OU) model is the best fit by a significant margin. An OU model portrays gradual adaptive evolution towards a new optimal state (here

caused by an abrupt change in local environmental conditions for the colonizers). A data set where trait changes could be explained by plastic responses only would not fit the OU model.

In the first phase of the OU model, the population has not yet reached the optimal state and is evolving toward the adaptive peak. It often takes evolution several generations to reach a new peak on the adaptive landscape (see Hunt et al. 2008 for an example). Gradual and directional trait change is accordingly expected in this phase of the model as the trait evolves closer and closer towards the optimal state. In our data, reaching the new optimum took about 7 years. A purely plastic response would on the other hand not show this gradual change in the trait. In fact, if the trait change was solely due to plasticity, the new population would immediately have reached the new optimal trait value as the reaction norm for the trait in the founding population had to cover the optimal trait value. However, such a pattern of trait change cannot be described by an OU model.

In the second phase of the OU model, the population is close to the optimum, and stabilizing selection maintains the phenotype despite deviations that may occur due to drift, minor changes in the fitness optimum, additional direct and indirect selection operating on the trait, or plasticity. The latter factors were not explicitly mentioned in the first version, but we have included it in the revision. When such deviations occur, in either direction, the phenotype is apparently rapidly pulled back towards the optimum in at least three instances in our data set. Hence, although plasticity may occasionally cause deviations it is unlikely to explain these consistent pullbacks to the optimum (subsequent increases in body mass when annual body mass $< \theta$, and subsequent decreases in body mass when annual body mass $> \theta$). In any case, we have estimated that only 13% of the variance around the optimum in the stationary phase (the second phase in the OU model) remains to be explained by factors such as plasticity and drift after controlling for estimation uncertainty in sample means (Supplementary Table 2). This suggests that if any, the role of plasticity is modest in the phenotypic changes we describe here. We have incorporated these arguments in the text and discuss the possible role of plasticity in our study (pages 4-5, lines 86-95). Further, we provide a more detailed explanation of the assumptions and consequences linked to the OU model fitting (pages 3-4, lines 66-70). We think this should clarify this issue raised by the reviewer.

Second, the parameters of the OU model suggest that the evolutionary changes we observe correspond to a certain range of effective population sizes linked to a trait with heritability ranging from 10% to 70%. We validate these estimates by an independent molecular estimate of effective population size, which suggest that the type of change we observe roughly corresponds to a trait with 50% heritability, which is neither an entirely plastic nor entirely genetic trait. It is important to also consider the fact that no

morphological quantitative trait in any organism has ever been found to be 100% heritable in any published studies, and that all quantitative traits have a plastic component, whether it is body mass, body size, tarsus length or other skeletal measures. In a recent study monitoring 10 bird species (Teplitsky et al. 2014 PLoS ONE), body mass in the great reed warbler was found to be significantly heritable (21.5%) as in all the other nine passerine species, and to harbor a quite substantial amount of additive genetic variation (as measured with evolvability *sensu* Hansen and Houle 2008 Journal of Evolutionary Biology). Interestingly, body mass was also under negative selection and responded significantly over the study period (several years to decades). This suggests that body mass is indeed a highly evolvable trait in birds. We have incorporated this information in the text (pages 5-6, lines 116-118).

Last, we present data from annual mid-parent and offspring body mass, which reveals they are significantly correlated (slope: 0.597; $P=0.000204$) even when year is included as a covariate in the linear model (slope: 0.54; $P=0.004$; year: $P>0.5$) This supports the assumption that body mass has a significant heritable component also in our population (Supplementary Fig. 2).

Reviewer's comments:

I also wonder why the authors focused on variation in body size, which is known to have an important environmental component, rather than on size-dependent morphological traits known to have higher heritability, like the tarsus.

Response:

We chose to focus on body mass in this study because it is an important trait related to fitness in birds, especially in the context of climate-induced changes, which we hypothesize to be a main driver of the observed changes given the location at the southern range of the species' distribution. In the revision we have also included a discussion of a study on several bird populations, including a European reed warbler population, reporting a decrease in body mass over several decades in relation with increasing temperatures (Salewsky et al. 2009 Oecologia). Interestingly, the authors demonstrate that these changes are not consistent with phenotypic plasticity, but find support for microevolutionary changes.

Reviewer's comments:

A second concern is the use of the proportion of breeders as a measure of mean population fitness. While this can be criticized, it also suggests an alternative interpretation of the results: density-dependence leading to

individuals in poor condition or poorly developed. Note that this would also be well-described by an OU evolutionary model. The authors can argue that the finding that recaptured individuals tended to have smaller body mass goes against this interpretation. Given that the authors examine apparent rather than actual survival, however, this later finding can reflect that the habitat is sub-optimal (the main breeding site only contained 5-8 pairs) and hence is avoided by dominant, larger individuals. If they disagree with these interpretations, the authors should discuss why these alternatives are unlikely.

Response:

We understand the reviewer's point. However, we think this alternative hypothesis is unlikely for several reasons. First, mean population fitness is here measured by estimating the proportion of adults that breed each year (which are then assigned the maximum fitness) compared to the total number of individuals present in the population, including adults which are not breeding (which are then assigned a fitness of 0). This measure does not reflect the density of the population *per se*, and it is important to note that even at the time of colonization, the population consisted of a considerable number of individuals (N=13 one year after colonization) although only few adults were breeding, and has remained quite stable at around 25-30 adults since then. In contrast, the proportion of breeding adults has increased. Hence, a scenario where density would affect resource availability and lead to changes in body size seems unlikely.

Further, as pointed out by the reviewer, our recapture data supports a scenario where natural selection on body mass is driving evolution, and migrating individuals, which may occasionally visit the population outside the breeding season, were not included in the analysis. We briefly discuss these points in the revision (page 9, lines 185-186).

Finally, we have included a new analysis in the supplementary material, where we show that mean annual variation in body mass (as measured by standard deviation (SD)) decreases significantly with mean population fitness, which strongly supports the hypothesis of natural selection on body mass over other alternatives (Supplementary Fig. 1).

Reviewer's comments:

Finally, I miss a convincing functional explanation linking body mass with fitness. The surface-volume ratio seems unconvincing to me because I'm not

aware of any study demonstrating that this ratio affects survival in sub-tropical environments. But I could be wrong here. Please add evidence in birds suggesting that this can be an explanation for the observed decline in body size.

Response:

As explained above, in reed warblers, changes in body mass have previously been reported in conjunction with rising temperatures (Salewsky et al. 2009 *Oecologia*), which is in accordance with Bergmann's rule for ectothermic species. In the latter study the authors find evidence not only for a correlation between increasing temperatures and lowered body mass but also with shorter feather length, which strongly point towards a thermoregulation hypothesis, even in a temperate European climate (see changes pages 6-7, lines 140-142). We now also refer to a study demonstrating the importance of surface-volume ratio for birds in response to varying temperature. Indeed, large bodied individuals start to lose water above the basal rate at lower temperatures than small bodied individuals (Gardner et al. 2011 *Trends in Ecology and Evolution*).

In addition, temperature is usually the strongest environmental correlate with body size in birds (Olson et al. 2009 *Ecology Letters*) and such physiological scaling has been documented in temperate and tropical climates (Ramirez et al. 2008 *Global Ecology and Biogeography*). Finally, we now also discuss a recent study (Eglington et al. 2015 *Global Ecology and Biogeography*) singling out the reed warbler as a species which is very sensitive to global warming, which has resulted in a recent range shift northwards. This clearly suggests that local temperature in the island of Malta is likely to have imposed a strong selective pressure on colonizers.

Reviewer's comments:

In sum, I think the authors have made an excellent job in conducting the analyses yet they need to provide additional evidence to convince the reader that what they report is an unprecedented evolutionary response in an extremely small population. If they can do so, the study will be an important contribution to evolutionary biology, with important implications for biological invasions (the role of evolution in invasion success is often under-estimated) and conservation biology (as demonstration of evolutionary rescue).

Response:

We thank the reviewer for these positive and insightful comments, and with the changes detailed above we do think we present a convincing case of an evolutionary response in a very small population.

Reviewer #2 (Remarks to the Author):**Reviewer's comments:**

*This manuscript describes the evolution of a very small population of reed warblers (*Acrocephalus scirpaceus*) following the colonization of a newly created wetland. The authors elegantly and clearly demonstrate evolution in body size using multiple approaches over only two decades after colonization. They further support a clear Ornstein-Uhlenbeck process, where the population shifts to a new adaptive peak and then shows variation around the new optimum. The authors claim that the results are particularly important given the relatively rapid rates they demonstrate in a population of less than 50 individuals (with an appreciable level of inbreeding), and they briefly discuss the implications for conservation biology more broadly.*

Overall, this is a very interesting study with valuable results and implications for the broader field of evolutionary conservation biology. Contemporary evolutionary processes have only begun to be considered seriously for conservation within the last decade, and empirical results like this one are key for both understanding the implications of contemporary evolution and in demonstrating the need of conservation practitioners to consider it.

Response:

We thank the reviewer for seeing merit in our study. We have re-centered some of the discussion around the benefits of our study to conservation biology (especially in the context of range shifts as mentioned below), as we feel that it is indeed a crucial feature of this manuscript (see Introduction and Discussion).

Reviewer's comments:

I have a few reservations about the interpretations of the result and the overall pitch of the manuscript, however:

1. The link to restoration is unclear (and probably unnecessary). Can you justify why restored habitats are an evolutionary challenge to species of conservation concern? The stated purpose of restoration is to assist population numbers and vital rates of native flora and fauna, so it is unclear why these environments would be a challenge to species due to strong changes in selective forces. I think there is a case to be made (e.g., restoration focusing on ecosystem traits that attract colonists, while not maintaining the correlations with ecosystem services that those habitat characteristics usually indicate), but it is not clear as written, and seems, in fact, opposite what one would expect. Further the hypothesis you propose to explain the change in selection pressure (thermoregulation) is one related to large-scale geographic location, not to anything to do with the restoration. As such, it is very unclear to me how the restoration itself is related to the larger story you are trying to tell. I think it might be simpler to merely frame the entire manuscript as "range expansion" or "colonizing a novel patch".

Response:

We understand the reviewer's point and thank him/her for these suggestions. The challenge facing this colonizing species was not entirely clear in the previous version of the manuscript. Indeed, the habitat has been restored with the aim of an assisted colonization process, and hence this habitat fits well with the needs of reed warblers, except for one crucial fact: it is located at the extreme southern edge of its current distribution. This should impose a strong selective pressure on colonizers in terms of high temperatures and lack of precipitation during the breeding season. We have accordingly revised the introduction and discussion with a stronger focus on the challenges of range expansions into extreme habitats.

Reviewer's comments:

2. The definition of survival may be problematic. How many of the recaptures were on the same year versus a subsequent breeding season? How high is survival on the breeding grounds? Presumably the majority of mortality occurs during migration in this species, so by defining mortality

the way you have here you have dramatically increased the probability that the "not recaptured" individuals are actually passage birds or floaters that never set up stable territories on Malta. In effect then your comparison between "recaptured" and "not recaptured" becomes a comparison between birds in the population and outside of it.

Response:

We disagree with this interpretation for several reasons. Although mortality is probably high during migration, we can see no *a priori* reason to assume that reproductive effort during the breeding season does not affect survival. Indeed, such a relationship has been documented in a number of bird species. Hence, although our recapture data may not account for total lifetime survival, it likely reflects a significant component of individual fitness. Most importantly, however, all birds included in this analysis were resident individuals captured during the breeding season when no migration occurs. Outside of the breeding season occasional migrants are sometimes captured on site, and they stay for a couple of days at most (Sultana et al. Breeding Birds of Malta 2011). We have included this crucial information in the revised version of the manuscript (page 9, lines 185-186).

Reviewer's comments:

3. All of the results can be explained as evolution due to immigration and emigration. It seems possible that the evolutionary change you document here could be the result of biased colonization by body size. This colonization bias could occur in conjunction with selection at the site (e.g., small bodied birds have higher fitness on site, but are less likely to get there due to correlations between body size and dispersal rate, so they fill in and outcompete larger bodied birds later), but colonization bias alone could also explain the results (e.g., the distribution of birds in the stable phase of the O-U process more closely represent the distribution of birds from the regional colonizing pool, but it took time to recreate that distribution because there are biases in colonizing potential that are associated with body size - larger bodied birds still get there first). This latter potential mechanism seems to be supported by the apparent increases in body size variance during the stable phase (doubly so, since the population size is actually larger at this point and that should make the SE appear to do the opposite?). But if you were to report data on the

distribution of body sizes elsewhere in southern Europe, this might assist in this regard.

Response:

We understand this concern, and as the reviewer suggests, we now report crucial new data in the supplementary material regarding the average body mass of migrating individuals which breed in other populations (presumably in Italy according to Procházka et al. 2008 Journal of Ornithology) but occasionally stopover in Malta and are thus likely part of the original colonizing pool (Supplementary Fig. 3). These individuals have a significantly higher body mass than the current Maltese population. Their body mass is in fact very close to the initial body mass of the population in 1996 (log body mass = 2.64 for individuals in Malta in 1996 vs. log body mass = 2.67 for individuals from other populations (N=30 sampled from 1997 to 2014) and log body mass = 2.43 for individuals in Malta in 2014). This fits very well with our hypothesis that large individuals colonized Malta in 1992 and onwards and that evolution of body mass occurred *in situ*.

Furthermore, variation in body mass variance (Fig. 1c) is mainly driven by fluctuations in sample sizes from year to year (which also influences SE estimates). We performed a linear regression on standard deviation (SD) in body mass against year, to formally test for changes in body mass variance over time, and this was not significant ($R^2=0.04$; $P=0.453$). Hence, the mechanism proposed by the reviewer does not have any support in that regard.

Reviewer's comments:

Your inbreeding coefficient gives some evidence to the contrary (i.e., if the population has a heterozygote deficit, that is unlikely to be produced entirely by immigration, assuming that the source populations are larger than this one ... which is reasonable), but it is hard to reject the fact that inbreeding might be much higher if it were not for new immigrants (can you calculate expected FIS for a population of that size and see how far off you are?). This is a very small population with high juvenile return rates. Further, your effective population size is three times the size estimated via field observations, which argues that both your genetic and evolution models are sampling part of a larger metapopulation that includes immigrants (given the size of the wetland, would it be fair to say that it is unlikely there could have been three times as many birds with stable territories on average each year than what

you detected?). Do you have good rates of immigration? Your observation on the speed of colonization would suggest that it is potentially high. Regardless, and most importantly, the fact that evolution is being driven by imm/emigration processes and not selection on site does not negate your findings in any way. It can still be measured as "selection", it is just selection due to colonization bias. The same O-U evolutionary processes can be invoked. It's a range-edge phenomenon vaguely akin to what has been reported in cane toads in Australia and holm oak in Europe. I just think it is critically important that you discuss how it is not possible to differentiate between the two possible pathways. Currently much of your language implies on-site selection.

Response:

We agree with the reviewer that our current level of inbreeding is so high that it does not fit a process through which immigration drives the evolutionary pattern observed. Our estimates of effective population size are accompanied with a certain level of uncertainty as reported in Table 2. The three independent methods of estimating N_e overlap, which means there are no significant differences between these estimates. Hence, there is no statistical support for the molecular estimates being higher than the field estimates. A meta-population type of scenario as suggested by the referee is therefore entirely speculative and lacks empirical support. We have therefore chosen not to discuss this alternative hypothesis in the text.

Surely, however, some occasional immigration must occur, and may actually add to the phenotypic deviations from the optimum observed during the late stages of the stationary phases. But, it is unlikely to explain our findings regarding selection on body mass and its correlation with mean population fitness. However, we provide a new analysis regarding our F_{IS} estimate, where we compare our estimate to F_{IS} estimates for the same set of markers in other European populations of reed warblers from a previously published study (Procházka et al. 2011 Journal of Avian Biology) (Supplementary Table 1). The new analysis shows that the level of inbreeding in Malta is above the average F_{IS} generally measured for this species (0.123 for Malta vs. a European average of 0.05 (SD=0.04); $P < 0.001$).

Reviewer's comments:

More minor comments

Ln 106 Can you estimate this over different time frames to see if selection is consistent despite slowing evolutionary rates?

Response:

This is an interesting point. To investigate whether there is any temporal variation in selection, we have conducted a linear model of recapture probability with both body mass and year of capture were included as covariates. Although body mass remains significantly correlated with recapture rate in this model ($P=0.007$), there is no significant year effect ($P=0.588$). Hence, selection remains fairly constant over time, despite slowing evolutionary rates. We have included this information in the revised version of the manuscript (page 5, lines 108-113).

Reviewer's comments:

Ln 112 Please admit that many other mechanisms are possible

Response:

We agree with the reviewer, and have added a discussion on other mechanisms that can affect our results (page 6, lines 138-140).

Reviewer's comments:

Ln 116 Under your model the fluctuations should be a combination of drift and stabilizing selection, if you are really at an "optimum". Given the SE you show in Fig. 1, it seems as though the population spans the projected "optimum" each generation, so I have no issue with assuming neutral net selection on average and looking for drift (although it would be good to be explicit about this assumption, as it sounds like you are claiming there is actually no selection at that point) to calculate the effective population size. But more interestingly, can you detect the reversed direction of selection if you treat the population as two, those below and those above the optima? Given the strong selection you show in the "approaching limb" phase, and the strength of the peak attractor, it should be detectable, even if you have to pool across time steps to obtain sufficiently large populations on either side of the optimum during this phase.

Response:

We thank the reviewer for bringing up several interesting points. We have made changes in the text to make it clearer that stabilizing selection is indeed operating in the stationary phase. The strength of

selection is proportional to the distance from the optimal trait state, which is why the trait has a tendency to evolve back towards the optimum if it gets displaced from it. However, and as described in an earlier response above, we have made it clearer in the manuscript that a range of factors (including drift and plasticity) can be reasons for the trait deviations away from the optimum in the stationary phase. We cannot use the model or any of its output parameters to estimate selection only in the stationary phase. In the OU model, the estimated alpha (the pull parameter) is assumed homogenous for the whole time series. A tendency to move towards the optimum is therefore part of the model, and the strength of selection is independent of whether the trait is bigger or smaller than the optimal trait size. The only thing that determines the strength of selection is the degree of maladaptation in the trait, where larger deviations from the optimum lead to stronger selection.

Reviewer's comments:

Ln 160 Were the birds individually marked with unique color combinations?

I'm assuming so, but it would be good to mention.

Response:

This is a good point, and we have added the requested information (page 9, lines 186-187).

Reviewer's comments:

Ln 346 I think you should explain here how you defined survival, but see my comment above, I think this definition of survival may be problematic

Response:

We agree that this is important and have added a sentence about the limitations of using this measure as a proxy for survival.

Reviewer #3 (Remarks to the Author):

Reviewer's comments:

This manuscript documents rapid phenotypic change in a small population of reed warblers following the colonization of a restored marshland. Over a 19-year period, the mean body mass in this population declined, and variation in fitness was correlated with body mass, suggesting the observed

changes were adaptive. Because the phenotypic changes exhibit an initially rapid decline, and are then followed by a more stationary phase, the interpretation is that the pattern of change is consistent with an Ornstein-Uhlenbeck process of how a trait evolves towards a new optimum.

I was very excited to see these results in the context of a restoration project, as it places evolutionary biology within a conservation framework. Unfortunately, the conclusion that the observed phenotypic changes represent evolution are not supported. The study fails to directly test what proportion of the phenotypic variation is due to heritable vs. environmental causes.

Response:

We thank the reviewer for his/her positive comments, and note that he/she acknowledges the fact that our data is consistent with an OU process. For the reasons explained above (response to Reviewer 1), this indicates that the observed phenotypic changes indeed represent evolution, although other factors may also contribute. See our revised Results section (Page 5, lines 86-95 and Supplementary Figure 2).

Reviewer's comments:

Numerous other studies have documented similar patterns of phenotypic change in natural populations, but they have also used pedigree-based animal models to test for heritable changes. For example, Charmantier et al. 2008 Science documented similar rapid phenotypic change over a 47-year period, but concluded the change were due to adaptive plasticity, not evolution. Indeed, the results presented in Charmantier et al. 2008 are qualitatively very similar to those presented here (i.e. a rapid change followed by a leveling off).

Response:

We believe the reviewer is mistaken in this instance. After careful reexamination of the Charmantier paper, we do not find any indication that the plastic changes occurred very rapidly at first and then levelled off, i.e. similar to an OU process. On the contrary, the changes described in that study are rather characterized by part of a quadratic curve describing a stasis phase followed by a phase where a more abrupt change is observed, which indeed are consistent with plasticity, unlike the ones we document.

Reviewer's comments:

Furthermore, reviews of studies documenting contemporary phenotypic changes in natural populations conclude that most of the observed changes also reflect plasticity rather than evolutionary change (e.g. Hendry et al. 2008 Molecular Ecology; Merilä and Hendry 2014 Evolutionary Applications).

My suggestion would be to revise the interpretation of the results to acknowledge that the observed changes could be due to either evolution or plasticity. Presenting the results as adaptive phenotypic change, rather than adaptive evolution, does not make the observed changes any less dramatic or interesting. I would also like to see a more thorough discussion of the mechanisms that could drive changes in body mass. The manuscript suggests temperature may be the mechanism, but mass, as opposed to size, can be influenced by food availability. Lastly, from a conservation perspective, a discussion of how the limits to adaptive plasticity vs. adaptive evolution might impact the long-term viability of this population would be very useful

Response:

As explained above we have revised the manuscript so as to include a discussion on how plasticity may have affected our data. Plasticity alone is unlikely to generate a pattern that strongly fits an OU model. Rather, plasticity is likely to have contributed to the observed deviations from this fit, although the variation that remains to be explained is small (around 13%) (see page 7, lines 163-165).

In the revision we also discuss how some studies (mentioned by the reviewer) on contemporary evolution have pointed towards an important role of plasticity, while still finding evidence for adaptive evolution in many cases (Merilä and Hendry 2014 Evolutionary Applications), and how none of them have found evidence for phenotypic changes of a plastic nature fitting an OU model (pages 4-5, lines 92-95).

Finally, we agree with the reviewer that a more thorough discussion of the biological mechanisms behind the evolution of body mass was needed. We have included a discussion of this in the revision along with additional references. Specifically, we have added an alternative hypothesis (thermal effects on resource availability) for why body mass might be the target of selection in this habitat (page 6, lines 139-140).

Reviewers' comments:

Reviewer #2 (Remarks to the Author):

I maintain that this manuscript is a very interesting study with valuable results and implications for the broader field of evolutionary conservation biology. Contemporary evolutionary processes have only begun to be considered seriously for conservation within the last decade, and empirical results like this one are key for both understanding the implications of contemporary evolution and in demonstrating the need of conservation practitioners to consider it.

The authors have done a very thorough job of addressing the reviewer concerns. Many of the things I thought would not be addressable are no longer a concern for me. The addition of the two supplemental figures, in particular, strengthen the argument immensely. I would like to emphasize that I agree with the authors response to the other reviewers in regards to plasticity. While confusion of plasticity and adaptation is a common concern in field-based studies, I do not believe that plasticity alone can produce an OU process (the evolution of a new reaction norm can, but that's still adaptation). I completely agree with the authors' rebuttal of the other reviewers. In addition, their presentation of very high heritability estimates (for a QTL morphological trait) makes it clear that the system has the additive genetic variance necessary to pull off the speed of adaptation they present here.

I have only very minor suggestions beyond this.

Ln 21 change "miniscule" to "small"

Ln 32 I like the new spin here a lot better. This paragraph (and many of the new sections) could use a reread and some editing for style, but the logic is good.

Ln 63 insert "and" after "process".

Ln 94 I buy this argument and its inclusion is a good addition

Ln 102 Change "decreased with mean population fitness" to "decreased as mean population fitness increased", or otherwise change the wording to indicate that the relationship is negative, not positive. This is a great addition.

Ln 108 While I don't disagree with this statement, perhaps change it to "this evolutionary change has resulted in a more successful breeding population". I think making the link between adaptation and its conservation consequences is important, but the initial wording suggests some level of evolution "for the good of the species", which is a common misconception among American university students. I may, however, be oversensitive to this issue; I readily admit. On a more pragmatic level, it's unclear whether a higher proportion of breeders during a given year really does increase population viability. If fewer breeders produce the same amount of young as when there are more breeders, and breeder identity switches among years randomly (although I'm sure it doesn't based on your results), then there is a situation where an increase in breeders would not necessarily be "beneficial for the reed warbler population". We can't assess that without more information on vital rates. But again, this is minor if you change the wording a little.

Ln 110 change "with" to "where", but some of this information should be in the methods

and you should just report the results of this test here (again, good addition).

Ln 121 remove "also"

Ln 121 You do not report these methods in the methods section, and more importantly, you do not say whether you used molecular-based parentage (controlling for extra-pair fertilization) or whether the parentage was based entirely on nest attendance. It's fine either way, but if it's the latter, then this estimate of heritability is actually biased low, which would be impressive, as this is a very high estimate of heritability for a quantitative morphological trait.

Ln 123 Combined with your inbreeding and effective population size estimates, I am now convinced that this is not the result of biased immigration. Well done marshaling this added evidence. Perhaps move this sentence to the discussion, however, so you have mentioned all three points of evidence before you make this conclusion?

Ln 142 Please remove all mention of "feather length". The authors of the cited work used flight feather length as their proxy, which is unrelated to thermoregulation and more closely tied to flight efficiency and migration distance. In their words "longer wing length is associated with greater migration distances (e.g. Pérez-Tris and Tellerría 2001; Leisler and Winkler 2003; Fiedler 2005). Further, evidence is accumulating that migrant bird species reduce migration distances and/or the proportion of migrating individuals in populations in connection with global warming (Fiedler 2003; Gordo et al. 2007). Therefore, climate-mediated changes in migratory behaviour may be affecting the wing lengths of some of the species that we examined." However! For another study showing decreases in passerine body size with increasing temperature across North America songbirds to give your argument full, north-temperate support, see Van Buskirk, Mulvihill, and Leberman. 2010. *Oikos* 119:1047. Incidentally, they found increases in "feather length" too, but again, it was flight feathers.

Ln 182 Great last statement.

Sincerely,
Brian J. Olsen
University of Maine
USA

Reviewer #3 (Remarks to the Author):

This is a revised version of a manuscript documenting rapid changes in body mass of reed warblers. The results show that following the colonization of a wetland on Malta, over a 19-year period body mass declines and then stabilizes. The main conclusion is that these temporal changes follow a O-U evolutionary process.

While the authors acknowledge that while some of the changes in phenotypic variation could be due to drift and plasticity, they argue natural selection is the primary driver of this pattern because reduced body mass is correlated with higher survival and the probability of breeding.

This is a very solid study based on a long-term data set. Nevertheless, I have some concerns.

First, despite the arguments made by the authors, there is no formal test to partition the temporal changes in body mass into that which is heritable vs. environmental. The authors argue that body mass has a significant heritability and harbors abundant additive genetic variation, but only a pedigree based analysis or common garden breeding experiment can provide definitive evidence for evolutionary change.

While I agree with some of the author's claims in support of evolution, I disagree with others. Specifically, the claim that plasticity would produce a perfect adaptive response (as opposed to the observed pattern) is only one of many ways in which environmentally induced variation could alter the phenotype. The main point here is that body mass may have evolved or not, but without a formal test, any conclusion will be subject to doubt. I recommend the authors be more open to this possibility in their presentation.

Second, one point that is not clear in the presentation, but has important implications for the interpretation of the results, is the migratory/resident status of the birds in the study. Unlike Darwin's finches which are year-round residents to isolated islands, the reed warbler is a migratory species that leaves its breeding grounds and winters in sub-Saharan Africa. Yet, it was unclear whether the Malta population was completely made up of non-migratory resident birds, or if it is migratory like other populations. The manuscript should be very clear on this point.

If the birds are migratory, a critical point is what proportion of birds measured in any given year return to breed in subsequent years, and how these individuals were dealt with in the analyses. The authors state

that breeding individuals captured more than once had their average mass used in the analyses. I wanted to know more about these individuals, and if the same individual across years exhibit changes in body mass that mimic the population declines, or if body mass is largely invariant across years. In either case, such individual averages become highly problematic when making evolutionary inferences from time series data. Have the authors looked at such individual variation across years?

Furthermore, if evolution is really occurring in situ as claimed by the authors, it would seem that between years only the offspring of local parents would be included in the analysis, or some other attempt would be made to account for overlapping generations. Based on the current presentation it appears that in any given year the individuals sampled for body mass represent a mix of birds descendent from the original founders and some number of new foreign recruits. Is this correct? Or is every individual in Figure 1 descendent from the original founders? It appears only about 10% of offspring recruit back into the population, which, while high for a migratory passerine, still begs the question where the other recruits into the population are coming from.

It is suggested that the original founding population came from Italy, but if there are new migrants that arrive each year, the question remains as to where they came from. How do we know they are not coming from other southern populations that exhibit small body masses? Clarifications on these points would help tremendously.

Lastly, very little information is presented on the habitat and ecological conditions over the 19 year period. The assumption of the O-U model is that there is a single optimum that the population is evolving towards. The validity of this assumption is difficult to assess because we don't know if other ecological factors covarying with time or remain stable. Did the vegetation change dramatically? Did insect availability change over time? What were the temperatures during the different breeding seasons?

Reviewer #2 (Remarks to the Author):

Reviewer's comments:

I maintain that this manuscript is a very interesting study with valuable results and implications for the broader field of evolutionary conservation biology. Contemporary evolutionary processes have only begun to be considered seriously for conservation within the last decade, and empirical results like this one are key for both understanding the implications of contemporary evolution and in demonstrating the need of conservation practitioners to consider it.

The authors have done a very thorough job of addressing the reviewer concerns. Many of the things I thought would not be addressable are no longer a concern for me. The addition of the two supplemental figures, in particular, strengthen the argument immensely. I would like to emphasize that I agree with the authors response to the other reviewers in regards to plasticity. While confusion of plasticity and adaptation is a common concern in field-based studies, I do not believe that plasticity alone can produce an OU process (the evolution of a new reaction norm can, but that's still adaptation). I completely agree with the authors' rebuttal of the other reviewers. In addition, their presentation of very high heritability estimates (for a QTL morphological trait) makes it clear that the system has the additive genetic variance necessary to pull off the speed of adaptation they present here.

Response:

We thank the reviewer for these positive and supportive comments. We agree that our data strongly supports adaptive evolution. However, we also agree with the other reviewer and the editor that we cannot entirely reject the possibility of adaptive plasticity, and we have made this clear in the discussion.

Reviewer's comments:

I have only very minor suggestions beyond this.

Ln 21 change "miniscule" to "small"

Response:

We have changed the wording as the reviewer suggested (ln 22).

Ln 32 I like the new spin here a lot better. This paragraph (and many of the new sections) could use a reread and some editing for style, but the logic is good.

Response:

We thank the reviewer for this helpful comment, and we have now rewritten some sections to improve the language/style.

Ln 63 insert "and" after "process".

Response:

We have reformulated this sentence (ln 68).

Ln 94 I buy this argument and its inclusion is a good addition

Response:

We thank the reviewer for these kind words.

Ln 102 Change "decreased with mean population fitness" to "decreased as mean population fitness increased", or otherwise change the wording to indicate that the relationship is negative, not positive. This is a great addition.

Response:

We agree with the reviewer's suggestion, and have changed the wording accordingly (ln 108-109).

Ln 108 While I don't disagree with this statement, perhaps change it to "this evolutionary change has resulted in a more successful breeding population". I think making the link between adaptation and its conservation consequences is important, but the initial wording suggests some level of evolution "for the good of the species", which is a common misconception among American university students. I may, however, be oversensitive to this issue; I readily admit. On a more pragmatic level, it's unclear whether a higher proportion of breeders during a given year really does increase population viability. If fewer breeders produce the same amount of young as when there are more breeders, and breeder identity switches among years randomly (although I'm sure it doesn't based on your results), then there is a situation where an increase in breeders would not necessarily be "beneficial for the reed warbler population". We can't assess that without more information on vital rates.

But again, this is minor if you change the wording a little.

Response:

We agree that it is best to avoid this kind of misunderstanding, and have changed the wording as the reviewer suggested (ln 115).

Ln 110 change "with" to "where", but some of this information should be in the methods and you should just report the results of this test here (again, good addition).

Response:

We have changed the wording as the reviewer suggested, and moved this information to the methods section (lines 234-236).

Ln 121 remove "also"

Response:

We have removed “also” (ln 125).

Ln 121 You do not report these methods in the methods section, and more importantly, you do not say whether you used molecular-based parentage (controlling for extra-pair fertilization) or whether the parentage was based entirely on nest attendance. It's fine either way, but if it's the latter, then this estimate of heritability is actually biased low, which would be impressive, as this is a very high estimate of heritability for a quantitative morphological trait.

Response:

We have now added some more information regarding this aspect in the methods (lines 243-246), and we reiterate that this analysis is not built as a pedigree based heritability analysis, as we do not possess such data. The positive parent-offspring correlation suggests some heritable basis of the trait studied here, but should not be confused with a proper estimate of heritability.

Ln 123 Combined with your inbreeding and effective population size estimates, I am now convinced that this is not the result of biased immigration. Well done marshaling this added evidence. Perhaps move this sentence to the discussion, however, so you have mentioned all three points of evidence before you make this conclusion?

Response:

We thank the reviewer for this comment. We see his point, but have decided to keep the sentence where it was, as we feel it is less relevant to the discussion.

Ln 142 Please remove all mention of "feather length". The authors of the cited work used flight feather length as their proxy, which is unrelated to thermoregulation and more closely tied to flight efficiency and migration distance. In their words "longer wing length is associated with greater migration distances (e.g. Pérez-Tris and Tellerría 2001; Leisler and Winkler 2003; Fiedler 2005). Further, evidence is accumulating that migrant bird species reduce migration distances and/or the proportion of migrating individuals in populations in connection with global warming (Fiedler 2003; Gordo et al. 2007). Therefore, climate-mediated changes in migratory behaviour may be affecting the wing lengths of some of the species that we examined." However! For another study showing decreases in passerine body size with increasing temperature across North America songbirds to give your argument full, north-temperate support, see Van Buskirk, Mulvihill, and Leberman. 2010. Oikos 119:1047.

Incidentally, they found increases in "feather length" too, but again, it was flight feathers.

Response:

We agree with the reviewer and have removed all mention of “feather length” in the text, and instead we now refer to the study by Van Buskirk et al. 2010 (line 146).

Ln 182 Great last statement.

Response:

We thank the reviewer for his kind comment and have kept this last statement despite other modifications we made to the text.

Reviewer #3 (Remarks to the Author):

Reviewer's comments:

This is a revised version of a manuscript documenting rapid changes in body mass of reed warblers. The results show that following the colonization of a wetland on Malta, over a 19-year period body mass declines and then stabilizes. The main conclusion is that these temporal changes follow a O-U evolutionary process.

While the authors acknowledge that while some of the changes in phenotypic variation could be due to drift and plasticity, they argue natural selection is the primary driver of this pattern because reduced body mass is correlated with higher survival and the probability of breeding.

This is a very solid study based on a long-term data set. Nevertheless, I have some concerns.

First, despite the arguments made by the authors, there is no formal test to partition the temporal changes in body mass into that which is heritable vs. environmental. The authors argue that body mass has a significant heritability and harbors abundant additive genetic variation, but only a pedigree based analysis or common garden breeding experiment can provide definitive evidence for evolutionary change.

While I agree with some of the author's claims in support of evolution, I disagree with others. Specifically, the claim that plasticity would produce a perfect adaptive response (as opposed to the observed pattern) is only one of many ways in which environmentally induced variation could alter the phenotype. The main point here is that body mass may have evolved or not, but without a formal test, any conclusion will be subject to doubt. I recommend the authors be more open to this possibility in their presentation.

Response:

We thank the reviewer for these comments. We agree that although our results are highly suggestive, they are not definitive, and we cannot reject plasticity without a formal test. We have made several modifications to the text, and we have added a sentence which clearly states this fact (line 178-180). We think we now report our results with a very open mind to this possibility and hopes this will satisfy both the reviewer and the editor.

Reviewer's comments:

Second, one point that is not clear in the presentation, but has important implications for the interpretation of the results, is the migratory/resident status of the birds in the study. Unlike Darwin's finches which are year-round residents to isolated islands, the reed warbler is a migratory species that leaves its breeding grounds and winters in sub-Saharan Africa. Yet, it was unclear whether the Malta population was completely made up of non-migratory resident birds, or if it is migratory like other populations. The manuscript should be very clear on this on point.

If the birds are migratory, a critical point is what proportion of birds measured in any given year return to breed in subsequent years, and how these individuals were dealt with in the analyses. The authors state that breeding individuals captured more than once had their average mass used in the analyses. I wanted to know more about these individuals, and if the same individual across years exhibit changes in body mass that mimic the population declines, or if body mass is largely invariant across years. In either case, such individual averages become highly problematic when making evolutionary inferences from time series data. Have the authors looked at such individual variation across years?

Response:

Indeed, the birds studied in this population are all migratory, as all reed warblers are. As stated in the text, we averaged the body mass of recaptured individuals across sampling events, which is a common practice to increase the reliability of measurements. In total, 99 out of the

392 birds measured were measured several times. We understand the reviewer's concern regarding the eventual temporal trend in body mass within individuals. However, individual variation was small and most repeated measurements were taken during the same breeding season, less than 30 days apart. Overall, differences in body mass between any two measurements at any time was on average of -0.0268 g. With an average weight of roughly 12 g, this is equivalent to a 0.002% difference between measurements of the same individual, which is not significantly different from zero ($P=0.765$). This suggests high repeatability of our measurements. The body weight of recaptured individuals did not change significantly over time ($R=0.002$; $P=0.980$; $df:204$). We also tested if individuals measured between 1996 and 2002 (the period of rapid evolution of body mass) that were measured across years (more than 250 days apart; $N=13$) could have influenced the observed pattern, but this was not the case ($R=0.004$; $P=0.990$ $df=13$).

We have included this additional information in the manuscript to dissipate any potential doubts in the readers (lines 215-217).

Reviewer's comments:

Furthermore, if evolution is really occurring in situ as claimed by the authors, it would seem that between years only the offspring of local parents would be included in the analysis, or some other attempt would be made to account for overlapping generations. Based on the current presentation it appears that in any given year the individuals sampled for body mass represent a mix of birds descendent from the original founders and some number of new foreign recruits. Is this correct? Or is every individual in Figure 1 descendent from the original founders? It appears only about 10% of offspring recruit back into the population, which, while high for a migratory passerine, still begs the question where the other recruits into the population are coming from.

It is suggested that the original founding population came from Italy, but if there are new migrants that arrive each year, the question remains as to where they came from. How do we know they are not coming from other southern populations that exhibit small body masses? Clarifications on these points would help tremendously.

Response:

The reviewer is correct, but as already answered in the former round of revision, the few migrants captured have relatively larger body mass (comparable to the Maltese birds in the first year after colonization; see supplementary figure 3) and are hence unlikely to affect our results, as also pointed out by reviewer # 2: "*Combined with your inbreeding and effective population size estimates, I am now convinced that this is not the result of biased immigration. Well done marshaling this added evidence.*".

Reviewer's comments:

Lastly, very little information is presented on the habitat and ecological conditions over the 19 year period. The assumption of the O-U model is that there is a single optimum that the population is evolving towards. The validity of this assumption is difficult to assess because we don't know if other ecological factors covarying with time or remain stable. Did the vegetation change dramatically? Did insect availability change over time? What were the temperatures during the different breeding seasons?

Response:

The ecological conditions have remained stable to the best of our knowledge and we have added some information about this point in the text (lines 175-178).